# A New Treatment Opportunity for DIPG and Diffuse Midline Gliomas: 5-ALA Augmented Irradiation, the 5aai Regimen

**DOI:** 10.3390/brainsci10010051

**Published:** 2020-01-17

**Authors:** Richard E. Kast, Alex P. Michael, Iacopo Sardi, Terry C. Burns, Tim Heiland, Georg Karpel-Massler, Francois G. Kamar, Marc-Eric Halatsch

**Affiliations:** 1IIAIGC Study Center, 148 College Street, suite 202, Burlington, VT 05401, USA; 2Southern Illinois University School of Medicine, Division of Neurosurgery, PO Box 19638, Springfield, IL 62794, USA; amichael@siumed.edu; 3Neuro-Oncology Unit, Department of Pediatric Oncology, Meyer Children’s Hospital, viale Pieraccini, 50139 24 Florence, Italy; iacopo.sardi@meyer.it; 4Department of Neurologic Surgery, Mayo Clinic, Rochester, MN 55905, USA; burns.terry@mayo.edu; 5Department of Neurosurgery, Ulm University Hospital, Albert-Einstein-Allee 23, D-89081 Ulm, Germany; tim.heiland@uniklinik-ulm.de (T.H.); georg.karpel@uniklinik-ulm.de (G.K.-M.); marc-eric.halatsch@uniklinik-ulm.de (M.-E.H.); 6Clemenceau Medical Centre, Department of Hematology-Oncology, Lebanese American University, Byblos Lebanon, City Centre Bldg. Suite 3A, Avenue Nouvelle, P.O. Box 1076, Jounieh, Lebanon; francois@kamarclinic.com

**Keywords:** 5-aminolevulinic acid, diffuse intrinsic pontine glioma, diffuse midline gliomas, fluorescence, glioblastoma, irradiation, photodynamic, prognosis, protoporphyrin IX, reactive oxygen species

## Abstract

Prognosis for diffuse intrinsic pontine glioma (DIPG) and generally for diffuse midline gliomas (DMG) has only marginally improved over the last ~40 years despite dozens of chemotherapy and other therapeutic trials. The prognosis remains invariably fatal. We present here the rationale for a planned study of adding 5-aminolevulinic acid (5-ALA) to the current irradiation of DIPG or DMG: the 5aai regimen. In a series of recent papers, oral 5-ALA was shown to enhance standard therapeutic ionizing irradiation. 5-ALA is currently used in glioblastoma surgery to enable demarcation of overt tumor margins by virtue of selective uptake of 5-ALA by neoplastic cells and selective conversion to protoporphyrin IX (PpIX), which fluoresces after excitation by 410 nm (blue) light. 5-ALA is also useful in treating glioblastomas by virtue of PpIX’s transfer of energy to O_2_ molecules, producing a singlet oxygen that in turn oxidizes intracellular DNA, lipids, and proteins, resulting in selective malignant cell cytotoxicity. This is called photodynamic treatment (PDT). Shallow penetration of light required for PpIX excitation and resultant energy transfer to O_2_ and cytotoxicity results in the inaccessibility of central structures like the pons or thalamus to sufficient light. The recent demonstration that keV and MeV photons can also excite PpIX and generate singlet O_2_ allows for reconsideration of 5-ALA PDT for treating DMG and DIPG. 5-ALA has an eminently benign side effect profile in adults and children. A pilot study in DIPG/DMG of slow uptitration of 5-ALA prior to each standard irradiation session—the 5aai regimen—is warranted.

## 1. Introduction

Here, we present the rationale and background for the planning of a clinical trial in the treatment of diffuse intrinsic pontine glioma (DIPG) or diffuse midline gliomas (DMG), called the 5aai regimen. There has been only marginal improvement in the treatment of DIPG over the last 40 years, despite dozens of chemotherapy and radiotherapy clinical trials [1,2]. The lack of progress is accompanied by increased side effect burden and decreased quality of life for these patients. The 5aai regimen utilizes 5-aminolevulinic acid (5-ALA), a tumor fluorescing drug currently used in the resection of glioblastoma (GB), to augment standard therapeutic irradiation. 

The extreme difficulties of treating DIPG are well-known:Tumor location behind a blood–brain barrier limits drug delivery;The tumor mass resides within vital structures that cannot be severed;Wide spatial and temporal cellular heterogeneity limits what any one drug can do;Wide brain dissemination of malignant cells and extrapontine extension has already occurred at the time of first diagnosis;Difficulty of biopsy results in paucity of pathophysiological knowledge; andExtreme heterogeneity of cell populations within the tumor limit the information value of any one biopsy.

These difficulties also largely apply to DMGs. In the past, DIPG has been used to identify all gliomas arising from or the pons. In the 2016 World Health Organization (WHO) classification of central nervous system tumors, the more common malignant version of DIPG was reclassified as DMG, H3K27M due to the consistent discovery of K27 mutations in the histone H3 gene *H3F3A*, or less commonly in the related *HIST1H3B* gene. Although there are significant physiological and epidemiological differences, GB, DMG, and DIPG are all WHO Grade IV tumors of astrocytic origin with a highly malignant nature and a diffuse infiltrative growth pattern [3]. Pathological grade by the WHO 2007 criteria do not predict survival in H3K27M mutated gliomas [4]. Additionally, H&E histology does not predict survival in DIPPGs [5]. 

Histological analysis has revealed that DIPG is really a whole brain disease. Leptomeningeal spread and diffuse invasion of isolated DIPG cells or cell clusters throughout corticospinal tracts, the brainstem generally, and thalamus are common. Remarkably, spread even into the frontal lobes can be seen at autopsy [6]. 

Extraordinary data, heretofore pathophysiologically unexplained, indicate that it has also been found that human DIPGs xenotransplanted into mouse pons directly from fresh post mortem biopsies result in tumors of identical or very similar H&E histology to the human donating tissue, but bearing murine immunohistochemistry markers [6]. For effective treatment, malignant DIPG cells separated from the main MRI-enhancing mass must also be addressed as both are currently intractable.

Irradiation at ~1.25 MeV is the mainstay of current DIPG treatment [1,7]. After initial intensity-modulated conformal radiation of 54–60 Gy delivered in ~2.0 Gy fractions, DIPG median survival remained 12 to 16 months. A second, lower dose of irradiation is possible, but does not substantially change outcome [7,8]. MRI fiber tractography of corticospinal tracts show that re-irradiation can diminish DIPG infiltration, displacement, and tract disruption with corresponding improved temporary functioning, even when overall survival is not prolonged. 

In DIPG—and probably in most cancers generally—strong functional cooperativity exists (juxtacrine, paracrine, and endocrine) between different subclones within the same tumor [9]. This implies a requirement for a broad spectrum approach, irradiation, and/or polypharmacy. 

## 2. Selective Uptake of 5-ALA by Glioma Cells

5-ALA is a 131 Da naturally occurring, and pharmaceutically synthesized amino acid that readily crosses the blood–brain barrier. It is an FDA and EMA approved drug (generically available or proprietary as Gliolan^TM^ or Gleolan^TM^) that is marketed for use during GB and other cancer surgeries to aid gross total resection [10,11]. Glioma cells selectively take up 5-ALA in direct proportion to their malignancy grade [12,13], and it is this selectivity that is the core of what will make 5aai so useful in treating DIPG or DMG.

After intracellular uptake, 5-ALA is then used to synthesize intracellular protoporphyrin IX (PpIX). Tumor cells have a lower ferrochelatase activity. As ferrochelatase is a link in synthesizing heme from PpIX, it is one of the rate limiting steps. Therefore, the combined effect of preferential 5-ALA uptake by glioma cells and reduced diverting of PpIX to heme results in strong preferential accumulation of PpIX in high grade glioma cells compared to non-malignant cells [10,11,12].

Since PpIX fluoresces at ~635 nm (red) during illumination with ~410 nm (blue) light, visual intraoperative demarcation of dense glioma cell areas can be made, although there are always scattered PpIX fluorescing cells within the resection cavity wall that cannot be seen or surgically addressed in their entirety [13,14]. Maximum strong PpIX fluorescence occurs 7 to 8 hours post 5-ALA ingestion. Weak GB fluorescence occurs at 8 to 9 hours post-ingestion.

5-ALA also functions in photodynamic therapy (PDT) in that light activation at ~410 or 635 nm of the selectively accreted PpIX generates reactive oxygen (singlet oxygen molecules, **^1^**O**_2_** and others) that in turn causes cytotoxicity to glioma cells or any other cell that has preferentially taken up 5-ALA [15,16]. In PDT, photon energy is low, non-tissue self-destructs. PpIX acts as a transducer of photon energy, efficiently taking low energy photons at 410 or 635 nm and transferring that energy to O**_2_**. The so generated **^1^**O**_2_** is then tissue destructive.

The precise energy pathways that transform light and higher energy irradiation energy into reactive oxygen species (ROS) and other cytotoxic products are not precisely understood. There appear to be at least several mechanisms. The common element is an optical excitation of the photosensitizer, in our case PpIX, to a fluorescing state, but this excited state need not be reached directly by photon absorption. There are other potential pathways of energy transfer involving Compton scattering and radiationless de-excitation in PpIX that also lead to enhanced ROS generation. It seems that diverse modes of energy transfer to PpIX can be effective to generate ROS, and irradiation is among them.

Multiple pediatric central nervous system tumor cell lines (medulloblastoma, pediatric neuroendocrine tumors, atypical teratoid rhabdoid tumor, and ependymoma) exhibit preferential uptake of 5-ALA and consequent PpIX fluorescence [17,18,19,20]. In a clinical study of 78 pediatric CNS tumors, 85% of GBs and 60% of anaplastic astrocytomas fluoresced after preoperative 5-ALA, enough to be judged helpful to the surgeon [18]. Mouriuchi et al. demonstrated that areas of a DIPG also fluoresced intraoperatively after preoperative 5-ALA [19] as do anaplastic foci of diffuse gliomas generally [21].

## 3. 5-ALA Enhances keV and MeV Cytotoxicity

Interestingly, 5-ALA with subsequent intracellular PpIX accumulation also increases keV or MeV irradiation cytotoxicity in a variety of contexts [22,23,24,25,26,27,28,29,30,31,32,33]. A robust, detailed physics-based explanation for this is currently lacking, but ample preclinical data attest to the basic mechanism of PpIX interacting with keV photons to generate cytotoxicity greater than the same irradiation given without 5-ALA where 5-ALA serves as a radiosensitizer. These twelve recent studies drive the imperative to study 5-ALA during the irradiation of DIPG [22,23,24,25,26,27,28,29,30,31,32,33]. 

Examples:

Kaneko showed similar 5-ALA sensitizing effects to carbon ion (~1.4 GeV) irradiation [23]. This was in a similar energy range as the carbon ion irradiation used in glioblastoma treatment [34]. Ueta et al. [25] used a Cs 137 source (peak irradiation at ~700 kev) and confirmed past data showing that 5-ALA enhanced glioma cell cytotoxicity of standard irradiation. Ueta et al. further showed that 5-ALA accretion alone damaged glioma cell mitochondria, setting up the release of ROS.

Park et al. showed that intraperitoneal administration of 5-ALA six hours prior to 3 Gy irradiation given daily for 10 days (30 Gy total) to a patient-derived orthotopic GB xenograft resulted in 22% longer survival than the same irradiation without 5-ALA and 50% longer than the control xenografted mice [35].

Malignant cell radiosensitization by selective 5-ALA uptake was also seen in preclinical models of cancers other than glioma, for example, colon, melanoma, and prostate [22,24,26]. This radiosensitization is thought to occur through mitochondrial damage mediated by PpIX, a metabolic product of 5-ALA that captures light or keV or MeV photons and transfers the energy to O**_2_** to then generate singlet oxygen. 

In a 2016 study of 5-ALA-guided surgery to remove cerebral metastases, Kamp et al. showed a lower local recurrence rate in patients with 5-ALA fluorescent metastases as compared to 5-ALA non-fluorescent metastases, but only in those patients who received post-surgical adjuvant local irradiation. The lower local recurrence rate might be explained by residual intracellular PpIX in the malignant cells that augmented post-resection irradiation treatment [36]. The effect was most evident for non-small cell lung cancer metastases.

The current standard 5-ALA dose, 20 mg/kg three hours prior to surgery or PDT, might be lower than optimal. Doses of 50 mg/kg can be given prior to glioma surgery that are well-tolerated and result in stronger intraoperative fluorescence [37,38]. CTCAE grade 1 skin redness and peeling were seen at higher doses. We have indications from the studies above that irradiation cytotoxicity augmentation is proportional to quantitative intracellular PpIX accretion. 

Clinically, in human squamous cell carcinoma in situ, combining 3 Gy radiation with 5-ALA cured lesions that failed to respond to 5-ALA light photon PDT alone. This demonstrated a synergistic effect of the two treatment modalities: light 5-ALA PDT plus 3 Gy and 4 MeV electron irradiation [33]. 

The CAALA regimen, recently devised for improving the effectiveness of 5-ALA PDT in GB, can be adapted and applied to 5aai treatment of DIPG [39]. The CAALA regimen uses three repurposed, FDA and EMA approved non-oncology drugs to increase 5-ALA uptake and conversion to PpIX in glioma cells. CAALA uses the antibiotic ciprofloxacin, the iron chelator deferiprone, and xanthine oxidase inhibitor febuxostat, along with thymidylate synthase inhibitor 5-fluorouracil (5-FU), where each have individually documented the effect of increasing the selective uptake of 5-ALA and/or selective conversion to PpIX. Several of the CAALA drugs have preclinically demonstrated anti-glioma effects independent of 5-ALA or PDT related effects [40]. CAALA simply advocates using all four drugs during open, post-resection 5-ALA PDT. CAALA has not had a formal clinical trial in GB, so the risks, potential benefits, and applicability of CAALA to DIPG or DMG are unknown.

Given the recent return of interest in I-125 brachytherapy of GB [41,42,43,44,45], adding 5aai and associated CAALA elements to brachytherapy has the potential to increase brachytherapy’s therapeutic index. I-125 decay results in 35 keV photons. A typical I-125 dose (T½ 59 days) to gliomas has been delivered at a rate of 0.05 Gy/h, resulting in 100 Gy at 1 year and 104 Gy at infinity. Since radiation necrosis and radiation-related brain malfunction are related to dose, using lower doses resulting from increased selectivity from 5-ALA would be highly desirable. Published data on I-125 brachytherapy of pontine gliomas is sparse, with a median survival of 10 months in the few treated patients [42]. Likewise, few data are available for I-125 treatment of brainstem gliomas in adults [43,45]. 

Photofrin is a hydrated derivative of PpIX also used in the clinical PDT of malignancies. Photofrin, like 5-ALA and PpIX, is relatively nontoxic by itself, but becomes highly cytotoxic after illumination with 630 nm light. Photofrin also has an extensive database on enhancing x-ray cytotoxicity to cancer [46,47] 

## 4. Safety

5-ALA use in pediatric gliomas including GB is remarkably well-tolerated with transient minor liver enzyme elevations and occasional evanescent skin effects as the only side effects, the same side effect profile as seen in use in adults [48,49]. In adults, CTCAE grade 1 transient skin redness and peeling is progressively more common as the 5-ALA dose approaches the maximum clinically tested of 50 mg/kg [37]. Only asymptomatic and transient CTCAE grade 1 liver function enzyme elevations have been seen at this highest tested dose. Circulating PpIX has a half-life of 8 hours after 60 mg/kg oral 5-ALA in humans [50], thus, accumulation is unlikely with daily administration.

Nevertheless, the proposed pilot study of 5aai in DIPG must include close monitoring of hepatic and renal function prior to each 5-ALA administration.

## 5. Conclusions

That 5-ALA augments keV irradiation cytotoxicity to glioma cells in direct proportion to their malignancy grade works in our favor. We may expect better malignant cell cytotoxicity from standard irradiation in PpIX-loaded DIPG cells that lie outside the main field of standard irradiation as well as in the main tumor mass. If we can demonstrate clinically effective 5-ALA augmentation of irradiation effects on the primary DIPG mass, enlarging the radiation field may be required to kill those DIPG cells lying outside current standard irradiation fields, but at lower doses that would have less effect on non-affected brain tissues. 

DIPG has been one of the most treatment-resistant, intractable, and quickly lethal cancers known. Dramatic indicators such as “long term survivors” after relapse is currently defined as children who survive more than three months after relapse and where a second irradiation results in a median time to death of seven months favor a 5aai regimen trial. 

5-ALA augmentation of standard postoperative irradiation treatment of GB has not been clinically tried as yet. The same considerations as outlined in this paper for DIPG or adult DMG apply to GB, where a trial of 5aai would also be warranted but also with CAALA regimen augmentation.

Additionally, given that 5-ALA PDT can activate an otherwise quiescent local immune response under certain circumstances [51,52,53,54], 5aai has the potential to further our efforts to “harness synergistic biology between radiation and immunotherapy” [55].

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
