# Peer review of "A New Treatment Opportunity for DIPG and Diffuse Midline Gliomas: 5-ALA Augmented Irradiation, the 5aai Regimen"

_brainsci, 2020, doi:10.3390/brainsci10010051_

Round 1

Reviewer 1 Report

Your article titled "A New Treatment Opportunity for DIPG and diffuse midline gliomas: 5-ALA Augmented Irradiation, the 5aai Regimen" attempts to provide a new therapeutic opportunity in a patient population with dismal prognosis. Unfortunately, I have significant concerns related to the proposed mechanisms at the heart of the proposal. Standard radiotherapy for all intracranial neoplasms is delivered with megavoltage photons, outside of known 5-ALA excitation quoted within the paper. Superficial x-rays are currently only utilized for cutaneous/subcutaneous malignancies or intraoperatively. Your proposal late within the article notes I-125 brachytherapy seed implantation. This results in focal radiation necrosis in most clinical cases and prescription is typically to 5mm depth, thus only useful in gross total resection scenario and typically not performed at the brainstem. For these reasons, I have significant concerns with your articles proposal/rationale.

Author Response

Reviewer #1

Moderate English changes required.

REK- We cannot meet the reviewer’s request to edit language use without more specific comments. What exactly does the reviewer find in need of editing ? Native English speaking physicians have found this paper easy to understand and without grammar mistakes, typos or other language oddities. We have nevertheless rephrased several sentences throughout the manuscript to obviate any possible ambiguity.

Standard radiotherapy for all intracranial neoplasms is delivered with megavoltage photons, outside of known 5-ALA excitation quoted within the paper. Superficial x-rays are currently only utilized for cutaneous/subcutaneous malignancies or intraoperatively. 

We do state in the ms.l.. “Irradiation at ~ 1.25 MeV is the mainstay of current DIPG treatment [1, 7]. After initial intensity-modulated conformal radiation of 54 - 60 Gy delivered in ~ 2.0 Gy fractions, DIPG median survival remains 12 to 16 months.” Is that not clear enough ?

We do quote data on MeV irradiation being augmented by Cs 137 and other high voltage sources. “Ueta et al [25] using a Cs 137 source (peak irradiation at ~700 keV) confirmed past data showing that 5-ALA enhanced glioma cell cytotoxicity of standard irradiation. Ueta et al further showed that 5-ALA accretion alone damaged glioma cell mitochondria, setting up release of ROS.”

also”Clinically, in human squamous cell carcinoma in situ, combining 3 Gy radiation with 5-ALA cured lesions that failed to respond to 5-ALA light photon PDT alone. This demonstrated a synergistic effect of the two treatment modalities - light 5-ALA PDT plus 3 Gy, 4 MeV electron irradiation [33].“

We further clarified by amending wording…”Kaneko showed similar 5-ALA sensitizing effect to carbon ion (~1.4 GeV) irradiation  [23]. This was in a similar energy range as carbon ion irradiation used in glioblastoma treatment  [34].”

Your proposal late within the article notes I-125 brachytherapy seed implantation. This results in focal radiation necrosis in most clinical cases and prescription is typically to 5mm depth, thus only useful in gross total resection scenario and typically not performed at the brainstem. 

Your reviewer might confuse high energy cytotoxicity from I-131with low energy I-125. The main emission from I=125 is with low energy, 35 keV, photons as we stated in the ms. 

Reviewer 2 Report

 There is no data on how much 5-ALA is taken into DIPG or DMG.The question is whether side effects may be enhanced by administering 5-ALA before each radiation therapy, because the total dose will become large.

Author Response

Reviewer #2.

There is no data on how much 5-ALA is taken into DIPG or DMG.

Does the reviewer contest the case report of Moriuchi et al [our reference 19] ? In our references 20 and 21, we report data showing that foci of grade 4 histology tend to be PpIX fluorescence positive in pediatric brain tumors other than DIPG. Given that DIPGs typically do have such foci (by H&E evaluation) we can expect those areas of a DIPG to disproportionately take up 5-ALA. We heartily agree with the reviewer that this must be determined by pilot study.

The question is whether side effects may be enhanced by administering 5-ALA before each radiation therapy, because the total dose will become large.

Agreed. Indeed side effects, particularly elevation of liver function enzymes, may increase over time with repeated 5-ALA use. This is a clear risk of the 5aai protocol. Given that elevated liver function tests rapidly normalize after even high-dose [50 mg/kg] 5-ALA, we judge the risks worth studying, given the usual outcome of DIPG. Note that the ease of monitoring for any increased evidence of liver damage also increases safety margin of a pilot study of 5aai. We note the evidence of safety in repetitive PDT study of Eljamel [ref ] and the evanescent and benign side effect profile when using over twice the usual dose of 5-ALA in humans [our ref 37 and 38]. We have nevertheless added to the ms. a note of caution re. potential side effects of repetitive 5-ALA. “Nevertheless the proposed pilot study of 5aai in DIPG must include close monitoring of hepatic and renal function prior to each 5-ALA administration.“ and “ Only asymptomatic and transient CTCAE grade 1 liver function enzyme elevations were seen at this highest tested dose. Circulating PpIX has a half-life of 8 hours after 60 mg/kg oral 5-ALA in humans [48]. Thus accumulation is unlikely with daily administration.”